# NanoMethViz: An R/Bioconductor package for visualizing long-read methylation data

**Shian Su**[1,2]*, **Quentin Gouil**[1,2], **Marnie E. Blewitt**[1,2], **Dianne Cook**[3], **Peter F. Hickey**[1,2], **Matthew E. Ritchie**[1,2]*

**1** Epigenetics and Development Division, The Walter and Eliza Hall Institute of Medical Research, Melbourne, Australia, **2** Department of Medical Biology, The University of Melbourne, Melbourne, Australia, **3** Econometrics & Business Statistics, Monash University, Melbourne, Australia

* su.s@wehi.edu.au (SS); mritchie@wehi.edu.au (MER)

**Data Availability Statement:** Data is available within the software package at http://www.bioconductor.org/packages/release/bioc/html/

## Abstract

A key benefit of long-read nanopore sequencing technology is the ability to detect modified DNA bases, such as 5-methylcytosine. The lack of R/Bioconductor tools for the effective visualization of nanopore methylation profiles between samples from different experimental groups led us to develop the *NanoMethViz* R package. Our software can handle methylation output generated from a range of different methylation callers and manages large datasets using a compressed data format. To fully explore the methylation patterns in a dataset, *NanoMethViz* allows plotting of data at various resolutions. At the sample-level, we use dimensionality reduction to look at the relationships between methylation profiles in an unsupervised way. We visualize methylation profiles of classes of features such as genes or CpG islands by scaling them to relative positions and aggregating their profiles. At the finest resolution, we visualize methylation patterns across individual reads along the genome using the *spaghetti plot* and heatmaps, allowing users to explore particular genes or genomic regions of interest. In summary, our software makes the handling of methylation signal more convenient, expands upon the visualization options for nanopore data and works seamlessly with existing methylation analysis tools available in the Bioconductor project. Our software is available at https://bioconductor.org/packages/NanoMethViz.

## Author summary

Recently developed nanopore sequencing technology enables DNA methylation measurement on long DNA molecules. This technology provides a new tool for investigating DNA methylation, a form of DNA modification that plays an essential role in early development, and is linked to some forms of cancer through adulthood. There is a lack of R/Bioconductor software for effective visualization of methylation calls based on nanopore platforms, which hinders the analysis and presentation of results. We developed *NanoMethViz*, the first R package to create visualizations for nanopore methylation data at various summary resolutions. *NanoMethViz* produces publication-quality plots to inspect the broad differences in methylation profiles of different samples, the aggregated methylation profiles of classes of genomic features, and the methylation profiles of individual long

NanoMethViz.html and additional data is available at https://zenodo.org/record/4495921.

**Funding:** This work was supported by Australian National Health and Medical Research Council (NHMRC) (https://www.nhmrc.gov.au) Project grant 1098290 to MER and MEB, a Bellberry-Viertel (https://bellberry.com.au, http://viertel.org.au) Senior Medical Research Fellowship to MEB. The funders had no role in study design, data collection and analysis, decision to publish, or preparation of the manuscript.

**Competing interests:** The authors have declared that no competing interests exist.

reads. Our software provides an efficient data format for storing methylation information and converts data from popular methylation calling software to formats recognized by statistical methods available in the Bioconductor toolkit for further analysis. *NanoMethViz* allows researchers to more quickly and effectively analyze their data and produce high-quality figures to present their results.

This is a *PLOS Computational Biology* Software paper.

## Introduction

Recent advances from Oxford Nanopore Technologies (ONT) have enabled high-throughput, genome-wide long-read DNA methylation profiling using nanopore sequencers, without the need for bisulfite conversion [1, 2].

A common goal of genome-wide profiling of DNA methylation is to discover differentially methylated regions (DMRs) between experimental groups. There is currently no software in the R/Bioconductor collection [3] for easily creating plots of methylation profiles in genomic regions of interest from the output of popular ONT-based methylation callers. We have developed *NanoMethViz* to create visualizations that give high resolution insights into the data to allow visual inspection of regions identified as differentially methylated by statistical methods. This software has been developed for compatibility with other software in the Bioconductor ecosystem [3], allowing for access to a wealth of existing statistical and genomic analysis methods. Specifically, this provides compatibility with the comprehensive toolkit for representing and manipulating genomic regions provided by *GenomicRanges* [4], and the statistical methods for DMR analysis available in packages such as *bsseq* [5], *DSS* [6] and *edgeR* [7].

The size of the data produced by ONT based methylation callers is the primary challenge in creating plots within defined genomic regions. It is not feasible to load entire methylation data-sets into memory on a standard computer, and for regions spanning the average length of a human or mouse gene, there are often enough data points to make smoothing visualizations computationally prohibitive. Together, this makes the analysis of methylation data difficult without access to high-performance computing (HPC), restricting the accessibility of methylation research using ONT sequencers.

## Design and implementation

The *NanoMethViz* package provides conversion of data formats output by popular methylation callers *nanopolish* [5], *f5c* [8], and *Megalodon* into formats compatible with Bioconductor packages for DMR analysis.

At the time of writing, there is no consensus on the format for storing nanopore methylation data. The methylation callers *nanopolish*, *f5c* and *Megalodon* all produce slightly different outputs to represent similar information. Methylation calling from nanopore sequencing is still an active area of research and more formats are expected to arise. From the workflow presented in Fig 1A, *NanoMethViz* provides conversion functions from the output of various methylation callers into an intermediate format shown in Fig 1B, containing the minimal information for downstream processes. This intermediate format is used to create plots, and can be converted into various methylation count table formats and objects used by DMR detection functions using provided functions.

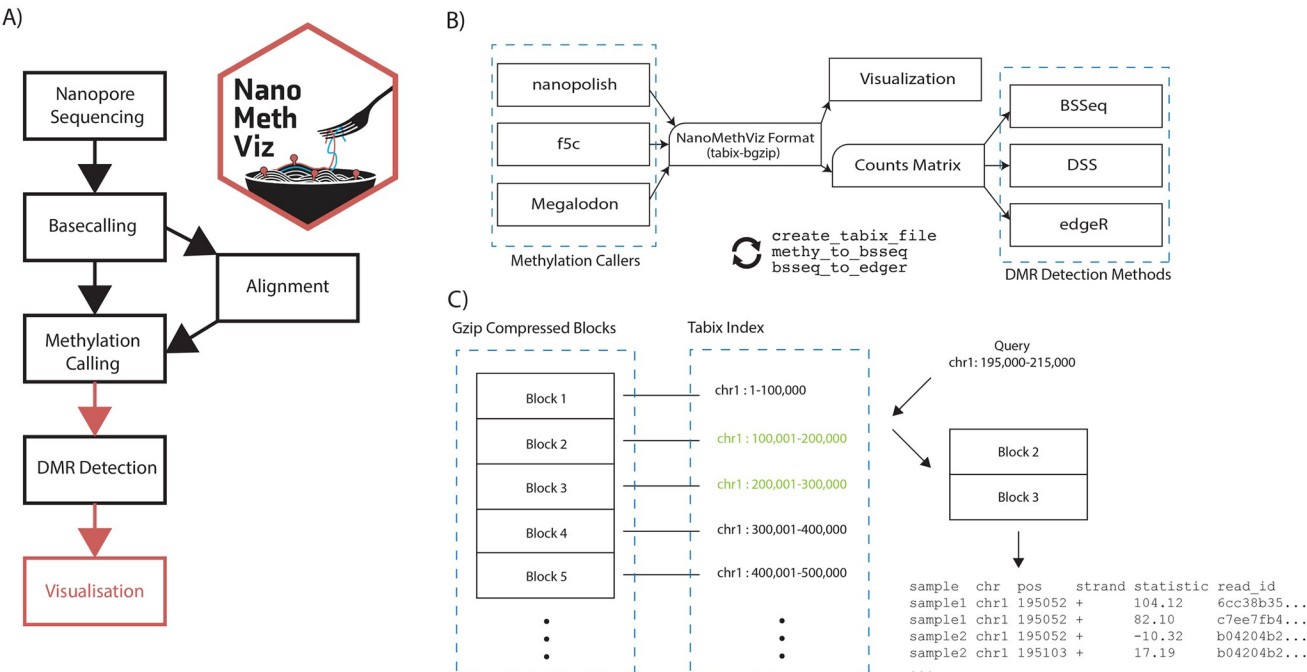

**Fig 1. Nanopore methylation workflow and data format.** A) The workflow used to perform differential methylation analysis. The red arrows indicate steps where further *NanoMethViz* provides conversion functions to bridge workflow steps. *NanoMethViz* performs visualization at the end of the workflow. B) Functions are provided in *NanoMethViz* to import the output of various methylation callers into a format used for visualization. This can be further converted by provided functions into formats suitable for various DMR detection methods provided in Bioconductor. C) The bgzip-tabix format compresses rows of tabular genomic information into blocks, and indexes the blocks with the range of genomic positions contained. This index is used for fast access the relevant blocks for decompression and reading.

*NanoMethViz* converts results from methylation caller into a tabular format containing the sample name, 1-based single nucleotide chromosome position, log-likelihood-ratio of methylation and read name. We choose log-likelihood of methylation as the statistic following the convention of *nanopolish*. This statistic can be converted to a methylation probability via the sigmoid transform as shown in Gigante *et al.* (2019) [9]. The intermediate format and importing functions provided by *NanoMethViz* enables compatibility with existing methylation callers, as well as simplifying extension of support for future methylation caller formats. The information contained in this format is sufficient to perform genome wide methylation analysis as well as retain the molecule identities that are an advantage of long reads.

As shown in Fig 1C, we compress the imported data using bgzip with tabix indexing. We use the tools *bgzip* and *tabix* included in *Rsamtools* toolkit [10, 11] to process the intermediate format; bgzip performs block-wise gzip compression such that individual blocks can be decompressed to retrieve data without decompressing the entire file, and tabix creates indices on position-sorted bgzip files to rapidly identify the blocks containing data within some genomic region. Having a format that is compressed with support for querying of data without loading in the whole data-set makes it feasible to analyse the data without the use of HPC, and allowing analysis to be performed on more widely available hardware.

Conversion is performed using block-wise streaming algorithms from the *readr* [12] package, this limits the amount of memory required to convert inputs of arbitrary size. Currently we support the import of methylation calls from *nanopolish*, *f5c* and *Megalodon*, and we also provide conversion functions from the tabix format into formats suitable for

differentially methylated region analysis using *bsseq*, *DSS* or *edgeR* using `methy_to_bs-seq` and `bsseq_to_edger`.

## Results

The primary plots provided by *NanoMethViz* are shown in Fig 2. They are the multidimensional scaling (MDS) plot and principal component analysis (PCA) plot for dimensionality reduced representation of differences in methylation profiles, the aggregate profile plot for methylation profiles of a set of features, and the *spaghetti plot* [9], for visualizing methylation profiles within specific genomic regions. While we have focused our development on 5mC methylation, in principle our work can be applied to any form of DNA or RNA modification.

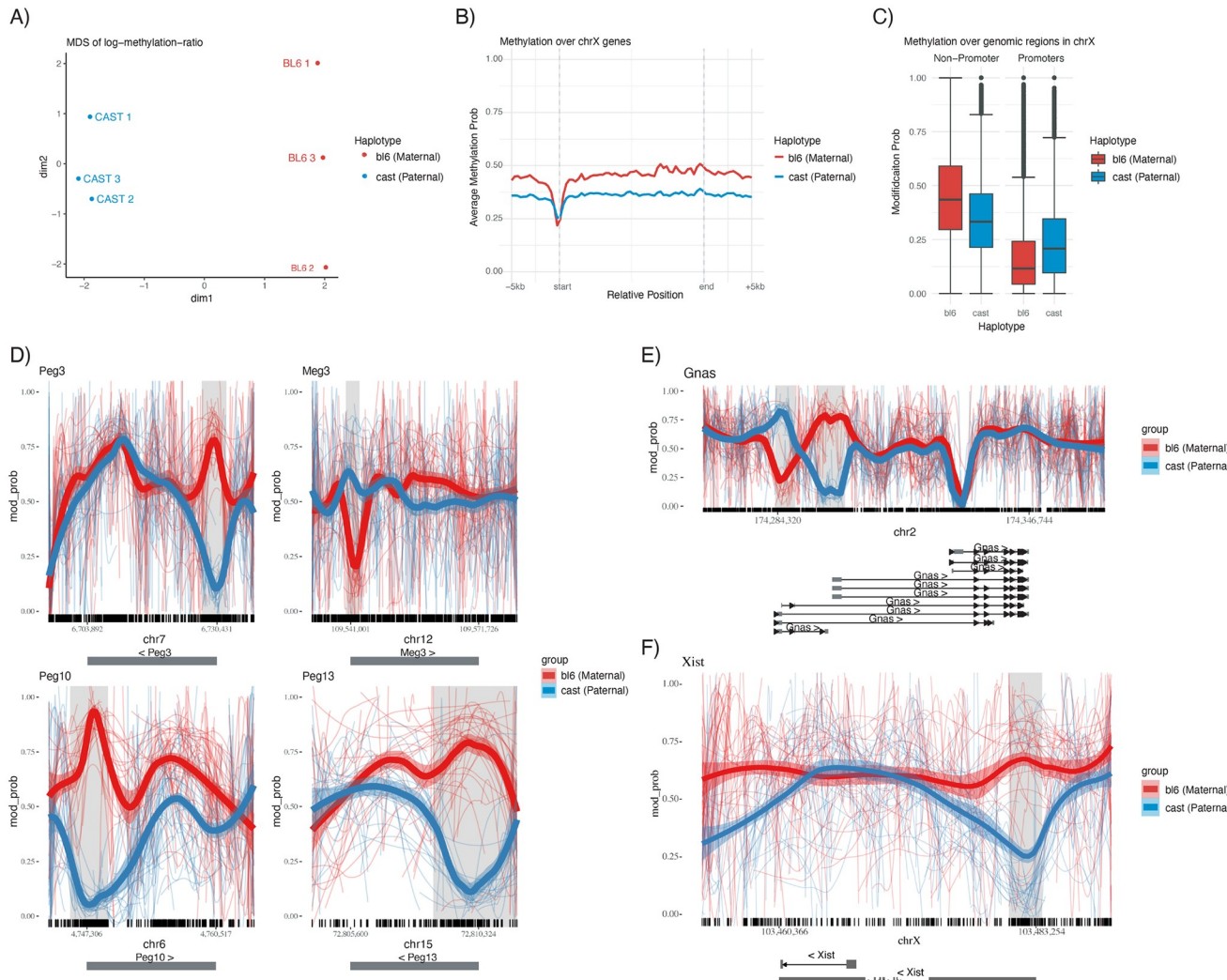

**Fig 2. Summary of the plotting capabilities of *NanoMethViz*.** A) Multidimensional scaling plot of haplotyped samples. B) Aggregated methylation profile across all genes in the X-chromosome, scaled to relative positions. C) Box plot of methylation probabilities over promoter and non-promoter regions for the BL6 and CAST haplotypes. D) Spaghetti plots of known imprinted genes *Peg3*, *Meg3*, *Peg10* and *Peg13*. Thin lines show the smoothed methylation probability on individual long reads, the thick lines show aggregated trend across the all the reads. The shaded regions are annotated as DMR by *bsseq*, and the tick marks along the x-axis show the location of CpG motifs. E) Spaghetti plot of *Gnas*, which shows two adjacent regions of opposite imprinting patterns. F) Spaghetti plot of *Xist*, a gene expressed from the inactive X chromosome.

We demonstrate the plots of *NanoMethViz* using a pilot dataset generated from triplicate female mouse placental tissues from F1 crosses between homozygous *C57BL/6J* mothers and *CAST/EiJ* fathers. Well characterized homozygous parents provided known SNPs for haplotyping reads [13], and the paternal X-chromosome is preferentially inactivated in female mouse placental tissue [14]. Together these two properties allow the parent of origin and X-inactivation state of each read to be known a priori when performing analysis of methylation profiles. The three samples of E14.5 placental tissue were harvested and each sequenced using a single PromethION flow cell, the data was basecalled using *Guppy* (v3.6.0) using the `dna_r9.4.1_450bps_hac_prom.cfg` high accuracy profile. Reads were aligned to the GRCm38 primary assembly obtained from GENCODE [15], using *minimap2* [16] (v2.16) with the ONT profile set by `-x map-ont` argument. The output of *minimap2* was sorted and indexed using *samtools* (v1.9), and only primary alignments were retained for analysis. The retained reads were haplotyped using *WhatsHap* [17] using mouse variant information provided by the Sanger Institute [13]. Methylation calling was performed by *f5c* [8] and associated with the haplotype information through the read IDs. *bsseq* (v1.26.0) [18] was used to identify differentially methylated regions and all visualizations in *NanoMethViz* were created using CRAN packages *ggplot2* [19] and *patchwork*.

The *MDS plot* shown in Fig 2A is commonly used in differential expression analysis to summarize the differences between samples in terms of their expression profiles. It represents high dimensional data in lower dimensions while retaining the high dimensional similarity between samples. We use the log-methylation-ratio to represent the methylation profiles of samples and provide the conversion function `bsseq_to_log_methy_ratio` to convert from a `BSseq` object to a matrix of log-methylation ratios. This matrix can be used with the `plotMDS` function from the *limma* [20] Bioconductor package to compute MDS components for the most variable sites following the *edgeR* bisulfite sequencing analysis workflow [21]. In Fig 2A, we see this approach shows separation of the haplotypes along the first dimension and according to sample (1,2,3) in the second dimension.

The *aggregation plot* shows aggregate methylation profiles across a class of features, revealing trends within a given class, such as promoters or repeat regions with fixed width flanking regions. It is produced by the function `plot_agg_regions`, which requires a table of genomic features or a `GRanges` object, and then plots the aggregate methylation profile scaled to the lengths of each feature such that they have the same start and end positions along the x-axis. The aggregation is an average of methylation profiles, with equal weights given to each feature as opposed to read, such that the aggregate is not biased towards features with higher coverage. This can be used to investigate specific classes of features such as genes or promoters. Fig 2B shows methylation profiles across all annotated genes in the X-chromosome, with the active X-chromosome (Xa) showing a higher level of methylation overall compared to the inactive X-chromosome (Xi). Genes from both chromosomes dip in methylation at the transcription start site, with Xi dipping below Xa by a small amount. This is further investigated in Fig 2C using the `query_methy` function to extract methylation data using ENSEMBL predicted promoters annotation to create a box plot. We see in the box plot higher levels of methylation in the maternal X-chromosome outside of promoter regions and lower levels of methylation within promoter regions. This matches previous observations in human fibroblast cells [22].

The *spaghetti plot* created by the functions `plot_region` or `plot_gene` visualize the methylation probability smoothed over experimental groups within specific genomic regions, as reported by methylation callers. The plot shows methylation probabilities smoothed along individual reads, annotations of CpG sites shown in tick marks along the x-axis, gene exons below the x-axis and top 500 most differentially methylated regions shaded in light grey.

Smoothing is performed over the methylation probabilities reported by methylation callers. A smoothed value near 0.5 can therefore arise either because adjacent CpGs have opposite methylation status (confidently called as 0.99 and 0.01) or because the caller has low confidence in the interval (probabilities around 0.5). Therefore biological and technical noise are confounded in the spaghetti representation. In Fig 2D the well known family of Peg and Meg genes are shown, which are paternally expressed genes and maternally expressed genes, respectively. In the case of paternally expressed genes *Peg3*, *Peg10* and *Peg13*, we see a drop in methylation in the paternal chromosomes near the TSS with an increase in methylation of the maternal chromosome. In the maternally expressed gene *Meg3* we see a drop in methylation in the maternal chromosome but a relatively small increase in methylation in the paternal chromosome. Fig 2E shows the methylation profile of Gnas, with two oppositely imprinted regions adjacent to each other. Fig 2F shows the gene *Xist*, which is expressed from the inactive paternal X-chromosome, we can see reduced methylation near the TSS of the gene on the inactive paternal chromosome. The spaghetti plots for individual reads allows visualization of methylation probabilities along single molecules; however, the data can appear noisy when plotted over large genomic regions, when coverage is high, and in regions with high site-to-site variation in methylation. In these placental samples, we see that there is a high level of variation in methylation probabilities outside of control regions and highly consistent signals within control regions. An alternative visualization for methylation along single molecules, where a heatmap of modification probability is plotted at each site, is implemented in *NanoMethViz* as `plot_region_heatmap` and `plot_gene_heatmap`.

The *aggregate plots* and *spaghetti plots* both use `geom_smooth` from *ggplot2* to create smoothed methylation profiles. Of the smoothing methods provided by `geom_smooth`, we found *loess* gave the most aesthetically pleasing fits. However, we found that *loess* scales poorly with the number of data points typically found in this type of data. To resolve this, the *spaghetti plot* takes per-site means before calling `geom_smooth` to significantly improve performance. In the *aggregation plot*, the methylation profiles are aggregated across the features, with relative positions within feature bodies and the two fixed width flanking regions without scaling. It was found that the feature region tends to have a much higher density of data points than flanking regions, leading to poor smoothing behavior as *loess* selects $N$ nearest points for fitting, with $N$ being a fixed portion of the total data. Many more points from the model fitting will be taken from the feature region than the flanking regions near the boundary between feature and flanking regions. To overcome this issue, we take binned means along the relative genomic positions, which results in data of uniform density along the x-axis. These optimizations allow smoothed plots of the genomic regions or aggregate features to be created where it would otherwise be infeasible by naive usage of the `geom_smooth` function.

## Discussion

The features provided by *NanoMethViz* fill current gaps in the data flow between software in the nanopore methylation analysis pipeline and the Bioconductor software ecosystem. The performance focused implementation of the plotting allows them to be generated without the need of high performance computers, facilitating more accessible analysis.

Other major software for visualization of long-read methylation data includes Python packages *pycoMeth* [23] and *methplotlib* [24]. *pycoMeth* provides a full workflow that produces a comprehensive interactive report on differentially methylated regions. *Methplotlib* is a plotting package for specified genomic regions with companion scripts for select analyses.

Both *pycoMeth* and *methplotlib* produce interactive plots of methylation data. *pycoMeth* produces summaries focused on CpG intervals, including a bar-plot with the count of

methylation intervals, a heatmap of the methylation status of CpG intervals, density plot of the methylation log-likelihood of significant intervals, and a karyoplot of the density of significant CpG intervals along the chromosomes. It also provides a higher resolution heatmap and density plot for significant intervals. The significance testing uses the Mann-Whitney U test for two samples or Kruskal-Wallis H test for three or more samples, with Benjamini and Hochberg correction for multiple testing. *Methplotlib* creates detailed plots of specific genomic regions, including a line plot of the methylation frequencies of individual samples, a heatmap of the methylation profiles on individual reads, and PCA as well as pairwise correlation plots for high-level inspection of data.

Compared with *pycoMeth*, *NanoMethViz* does not provide a complete pipeline for analysis; rather it is intended to be used as a modular component of a workflow that includes other Bioconductor software for a more flexible and powerful analysis. *NanoMethViz* contains conversion functions to import data from methylation callers into its standard format, then conversions from the standard format into formats appropriate for DMR callers from Bioconductor, including *bsseq*, *DSS* and *edgeR*.

*Methplotlib* is similar in operation to *NanoMethViz* when plotting genomic regions. *NanoMethViz* operates within interactive R sessions, as opposed to the command-line calls used by *methplotlib*. This allows the results of expensive operations such as annotation parsing to be kept in memory between plotting calls.

## Availability and future directions

The R/Bioconductor package *NanoMethViz* is available from https://bioconductor.org/packages/NanoMethViz, with all features shown in this paper available in the 2.0.0 release. Vignettes are provided with examples of how to import data from methylation callers and how to create the basic plots. Example data is included with the package including data from genes *Peg3*, *Meg3*, *Impact*, *Xist*, *Brca1* and *Brca2*. Data used for Fig 2A–2C can be found at https://zenodo.org/record/4495921.

In conclusion, *NanoMethViz* provides conversion functions, an efficient data storage format and a set of visualizations that allows the user to summarize their results at different resolutions. This work unlocks the potential for established Bioconductor DMR callers to be applied to data generated by ONT based methylation callers, lowers the hardware requirements for downstream analysis of the data, and provides key visualizations for understanding methylation patterns using ONT long reads.

Future development will support a wider range of plots, including some of those currently found in *pycoMeth* and *methplotlib* to make them available for R users. Ongoing support will be added for any new, popular methylation callers that arise with differing formats to existing callers.

## Acknowledgments

We thank Kathleen Zeglinski for designing the *NanoMethViz* logo and Kelsey Breslin and Tamara Beck for their assistance in generating the data used to test our software.

## Author Contributions

**Conceptualization:** Shian Su, Matthew E. Ritchie.

**Formal analysis:** Shian Su.

**Funding acquisition:** Marnie E. Blewitt, Matthew E. Ritchie.

**Methodology:** Quentin Gouil, Dianne Cook, Peter F. Hickey, Matthew E. Ritchie.

**Resources:** Marnie E. Blewitt.

**Software:** Shian Su.

**Supervision:** Quentin Gouil, Marnie E. Blewitt, Dianne Cook, Matthew E. Ritchie.

**Visualization:** Shian Su.

**Writing – original draft:** Shian Su, Matthew E. Ritchie.

**Writing – review & editing:** Shian Su, Quentin Gouil, Marnie E. Blewitt, Peter F. Hickey, Matthew E. Ritchie.

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
