## [Decision Letter · Decision Letter 0]

23 Jun 2021

Dear Mr Su,

Thank you very much for submitting your manuscript "NanoMethViz: an R/Bioconductor package for visualizing long-read methylation data" for consideration at PLOS Computational Biology. As with all papers reviewed by the journal, your manuscript was reviewed by members of the editorial board and by several independent reviewers. The reviewers appreciated the attention to an important topic. Based on the reviews, we are likely to accept this manuscript for publication, providing that you modify the manuscript according to the review recommendations.

Sincerely,

Dina Schneidman

Software Editor

PLOS Computational Biology

[LINK]

Reviewer's Responses to Questions

**Comments to the Authors:**

Reviewer #1: The authors have developed NanoMethViz, a tool for the visualization of modified nucleotides from nanopore data. I am quite familiar with this space, as I have developed a similar tool (methplotlib; doi.org/10.1093/bioinformatics/btaa093). The integration of NanoMethViz with the bioconductor ecosystem is highly valuable, especially in the context of further (statistical) analysis. Detecting nucleotide modifications from nanopore data is a fast moving field with many software tools and data formats, and there is clearly no one-size-fits-all approach. The use of NanoMethViz in an interactive R session is indeed valuable, compared to methplotlib which is purely command line based. The software appears well-documented and the visualizations of NanoMethViz are attractive and of high quality. As such NanoMethViz represents an important contribution to the field and wish to congratulate the authors with their work. The manuscript is well written and easy to follow.

Please find my specific comments below.

Sincerely,

Wouter De Coster

I am not unbiased in this matter, but I have my reservations about how NanoMethViz is compared to existing tools in this space. The 'Authors summary' states there is "a lack of software for effective visualization of methylation calls based on nanopore platforms", which is demonstrably incorrect. The introduction, however, is a more honest representation of the current state of the field, as the following is stated "There is currently no software in the R/Bioconductor collection for easily creating plots of methylation profiles". Some of the statements about methplotlib are also outdated, although these very well might have been correct at the time of writing the manuscript. Methplotlib similarly uses bgzip and tabix for efficient access to tabular files, and has PCA and pairwise correlation plots for a higher level assessment. Smoothening is done using a rolling window average. Interestingly enough, I also picked the GNAS locus as an example for my publication. There is absolutely no need for rivalry or competition between software developments and I consider NanoMethViz a valuable contribution to the field, but I would appreciate a more accurate representation (and citation) of tools with a similar scope.

I am a bit surprised about the noisiness of the individual reads (as shown in the spaghetti plots). Can the authors comment on this, and perhaps provide a recommendation towards which coverage would be minimally required for accurate average methylation probabilities?

Reviewer #2: In their manuscript “NanoMethViz: an R/Bioconductor package for visualizing long-read methylation data” Su et. al describe their newly developed Bioconductor package to visualize long-read methylation data. The package seems useful and is overall well described such that I would in principle recommend publication. Nevertheless, I have a few minor comments/suggestions that I think should be addressed:

1. Shouldn’t the methylation signal in single reads as for example displayed in the Spaghetti-Plots be binary i.e. 0 (Not methylated) or 1 (Methylated). I realize that methylation likelihoods or probabilities are displayed, but in that case shouldn’t values near 0.5 be excluded as unreliable? So, does it really make sense to smooth these values? Along these lines, assuming a binary signal for single CpGs along a read (thin lines), is a line-plot really the appropriate way to display this. Wouldn’t one rather use a point-graph (each CpG per read gets a point). This would avoid the appearance of continuity where there is none (weird zig-zagging). The smoothed regression line aggregating “everything” is fine, though, since here continuity is indeed intended (similarly to what is done in conventional short read DNA methylation analysis).

2. It is not clear how MDS can be performed without loading all of the data at once, a necessity to allow non HPC analysis as discussed by the authors. A bit more background on how this is achieved would be great.

3. Would it be possible to also enable PCA or even biplot visualizations? This would have the advantage of instantly getting an estimate of the main axis of variability as well as the driving molecular features (methylation patterns).

4. One more analysis/visualization to have (as a bonus) would be to calculate and visualize methylation correlations along the length of a read (i.e. correlation between two CpGs in dependence of their distance).

**Have the authors made all data and (if applicable) computational code underlying the findings in their manuscript fully available?**

Reviewer #1: Yes

Reviewer #2: Yes

PLOS authors have the option to publish the peer review history of their article (what does this mean?). If published, this will include your full peer review and any attached files.

Reviewer #1: **Yes: **Wouter De Coster

Reviewer #2: No

Figure Files:

Data Requirements:

Reproducibility:

References:

---

## [Decision Letter · Decision Letter 1]

4 Oct 2021

Dear Mr Su,

We are pleased to inform you that your manuscript 'NanoMethViz: an R/Bioconductor package for visualizing long-read methylation data' has been provisionally accepted for publication in PLOS Computational Biology.

Best regards,

Dina Schneidman

Software Editor

PLOS Computational Biology

Please cite methplotlib in the final version

Reviewer's Responses to Questions

**Comments to the Authors:**

Reviewer #1: I am satisfied by the response and adjustments from the authors to my comments and requests for clarification, and I have no further remarks. I wish to congratulate the authors on their manuscript and their welcome addition to the nanopore data visualization landscape.

Sincerely,

Wouter De Coster

Reviewer #2: All my comments have been addressed and I can now recommend publication.

Thank you for the nice tool.

**Have the authors made all data and (if applicable) computational code underlying the findings in their manuscript fully available?**

Reviewer #1: Yes

Reviewer #2: Yes

PLOS authors have the option to publish the peer review history of their article (what does this mean?). If published, this will include your full peer review and any attached files.

Reviewer #1: **Yes: **Wouter De Coster

Reviewer #2: No

---

## [Editor Report · Acceptance letter]

20 Oct 2021

PCOMPBIOL-D-21-00223R1 

NanoMethViz: an R/Bioconductor package for visualizing long-read methylation data

Dear Dr Su,

I am pleased to inform you that your manuscript has been formally accepted for publication in PLOS Computational Biology. Your manuscript is now with our production department and you will be notified of the publication date in due course.

With kind regards,

Katalin Szabo
